# Natural Mycoplasma Infection Reduces Expression of Pro-Inflammatory Cytokines in Response to Ovine Footrot Pathogens

**DOI:** 10.3390/ani12233235

**Published:** 2022-11-22

**Authors:** Adam M. Blanchard, Christina-Marie Baumbach, Jule K. Michler, Natalie D. Pickwell, Ceri E. Staley, Jemma M. Franklin, Sean R. Wattegedera, Gary Entrican, Sabine Tötemeyer

**Affiliations:** 1School of Veterinary Medicine and Science, University of Nottingham, Loughborough LE12 5RD, UK; 2Institute of Anatomy, Histology and Embryology, Faculty of Veterinary Medicine, Leipzig University, 04103 Leipzig, Germany; 3Moredun Research Institute, Pentlands Science Park, Bush Loan, Penicuik EH26 0PZ, UK; 4The Roslin Institute, The University of Edinburgh, Easter Bush, Roslin EH25 9RG, UK

**Keywords:** footrot, sheep, cell culture

## Abstract

**Simple Summary:**

Ovine footrot is a painful contagious disease of the hoof. Caused by the bacterium *Dichelobacter nodosus*, there is mounting evidence that other bacteria play an important role in the initiation of the lesions. We had previously established *Mycoplasma fermentans* as being highly associated with the disease and wanted to understand the immune-dampening effects on the local immune response to other bacteria associated with footrot. We created primary cell cultures of ovine skin cells from healthy foot biopsies collected at an abattoir. The initial cells were naturally infected with *Mycoplasma fermentans,* which, after isolation, were treated with antibiotics to create a Mycoplasma-free line. The different cultures were stimulated with different bacteria, and the mRNA and protein release were assessed under both conditions. The stimulation resulted in an increased expression of key immune indicators in the *M. fermentans*-free cells; however, this did not correspond to a protein release. The skin cells naturally infected with *M. fermentans* showed little response to stimulation. Therefore, we conclude Ovine skin cells infected with *M. fermentans* have a reduced response to stimulation dampening the immune response to other bacteria. This provides an important insight into the impact of multiple pathogens on the host response to footrot.

**Abstract:**

Ovine footrot is a complex multifactorial infectious disease, causing lameness in sheep with major welfare and economic consequences. *Dichelobacter nodosus* is the main causative bacterium; however, footrot is a polymicrobial disease with *Fusobacterium necrophorum*, *Mycoplasma fermentans* and *Porphyromonas asaccharolytica* also associated. There is limited understanding of the host response involved. The proinflammatory mediators, interleukin (IL)-1β and C-X-C Motif Chemokine Ligand 8 (CXCL8), have been shown to play a role in the early response to *D. nodosus* in dermal fibroblasts and interdigital skin explant models. To further understand the response of ovine skin to bacterial stimulation, and to build an understanding of the role of the cytokines and chemokines identified, primary ovine interdigital fibroblasts and keratinocytes were isolated, cultured and stimulated. The expression of mRNA and protein release of CXCL8 and IL-1β were measured after stimulation with LPS, *D. nodosus* or *F. necrophorum*, which resulted in increased transcript levels of IL-1β and CXCL8 in the *M. fermentans*-free cells. However, only an increase in the CXCL8 protein release was observed. No IL-1β protein release was detected, despite increases in IL-1β mRNA, suggesting the signal for intracellular pre-IL-1β processing may be lacking when culturing primary cells in isolation. The keratinocytes and fibroblasts naturally infected with *M. fermentans* showed little response to the LPS, a range of *D. nodosus* preparations or heat-inactivated *F. necrophorum*. Primary single cell culture models complement ex vivo organ culture models to study different aspects of the host response to *D. nodosus*. The ovine keratinocytes and fibroblasts infected with *M. fermentans* had a reduced response to the experimental bacterial stimulation. However, in the case of footrot where *Mycoplasma* spp. are associated with diseased feet, this natural infection gives important insights into the impact of multiple pathogens on the host response.

## 1. Introduction

Ovine footrot is a contagious disease characterised by the separation of the hoof from the underlying dermis of sheep and is the principal cause of lameness in UK flocks [1]. It is a significant welfare issue and has an enormous financial impact, which often limits interventions and treatments used to eradicate the infection from a flock which, in turn, reduces productivity [2]. With 90% of flocks in the UK affected, a mean prevalence of 4.5% [1], there is an increased pressure to reduce antibiotic use [3]. As a result, there is a drive to broaden our understanding of the disease and use novel experimental approaches [4,5,6,7] in an effort to improve the situation. Despite knowing the main causative bacterium, *Dichelobacter nodosus* (*D. nodosus*) [8], the risk factors involved, such as an impaired skin barrier and poor environmental conditions [9], and having vaccines available, little progress has been made to reduce the incidence. Additionally, the bacterium *Fusobacterium necrophorum* (*F. necrophorum*) has also been identified as a contributing factor to footrot. However, its role in the pathogenesis is unclear, with two putative hypotheses: *F. necrophorum* may facilitate disease development by damaging the interdigital skin and promoting interdigital dermatitis that subsequently permits the replication of *D. nodosus* [10,11]; or, secondly, *F. necrophorum* exacerbates the severity and persistence of footrot [8,9,12,13].

Recent publications have helped the understanding of the microbial communities present on the surface of the interdigital sheep skin [14,15] and deeper within the interdigital skin layers [5]. The bacterial species found deeper within the interdigital skin structure and associated with footrot were *D. nodosus*, *Mycoplasma fermentans,* (*M. fermentans)* and *Porphyromonas asaccharolytica* (*P. asaccharolytica)*. As *D. nodosus* often requires tissue damage and the corrosive action of other bacteria as a prerequisite for infection, the presence of *M. fermentans,* which has been shown to moderate the host immune response, may have an important role in disease susceptibility and progression.

The host response to the causative pathogen *D. nodosus* is not well understood and is complicated by footrot being a polymicrobial disease. It has been shown previously that, within outwardly appearing healthy interdigital tissues as well as footrot-affected tissues, the pro-inflammatory cytokines and chemokines interleukin (IL)-1β and C-X-C Motif Chemokine Ligand 8 (CXCL8), IL-6 and IL-17A showed a wide range of expression on the mRNA level, but no difference was observed in the context of footrot or healthy tissue [15]. An inflammatory cell infiltration scoring system also showed a wide range of inflammation in both healthy and footrot-affected interdigital skin [5]. A significant association between an elevated expression of IL-1β and *D. nodosus* load in the footrot samples, as well as correlations between the CXCL8 and IL-1β with *D. nodosus,* were only found in the footrot samples, but not in the healthy skin samples [15]. The epidermis consists of tightly packed keratinocytes producing antimicrobial peptides [16]. Once the basement membrane is overcome or even damaged, the fibroblasts, interspersed in the extracellular matrix of the underlying dermis, are activated and release pro-inflammatory cytokines, chemokines and prostanoids [16,17]. However, a comprehensive transcriptomic assessment of ovine interdigital skin identified the suppression of cytokines, chemokines, metalloproteases, and their regulators in the footrot-affected tissue, suggesting a local dampening of wound healing and immune cell recruitment [7]. Interdigital skin, composed of the epidermis and dermis, is at the forefront of the primary immune response against footrot.

The transcription of IL-1β, TNF-a and TLR2 has been shown to increase after stimulation with *D. nodosus* in a single-cell-type infection model using ovine fibroblasts [18]. Furthermore, the levels of IL-1β and CXCL8 were measured in a cell culture system using an interdigital skin explant model demonstrating the release of those proteins, and that they play a major role in the initial response to the presence of *D. nodosus* [19].

Here, we created primary cell cultures of ovine interdigital fibroblasts and keratinocytes to further understand the response of these skin cells to bacterial stimulation. To clarify the role of the cytokines and chemokines that were identified in an earlier explant model [19], the expression and release of IL-1β and CXCL8 were measured after stimulation with *E. coli* lipopolysaccharide (LPS), heat-inactivated *D. nodosus*, ultraviolet light-inactivated *D. nodosus*, formalin-fixed *D. nodosus* or heat-inactivated *F. necrophorum.* Due to the identification of a natural infection of ovine digital skin with *Mycoplasma fermentans* (*M. fermentans*) from metatranscriptomic data [7], these experiments were carried out with and without its presence in the ovine skin cell cultures.

## 2. Methods

### 2.1. Primary Cell Isolation and Culture

Healthy ovine distal limbs (*n* = 8) were collected from a local abattoir and cleaned thoroughly with brushes under running tap water, removing the adhering dirt to minimise bacterial and/or fungal infection. The interdigital area was disinfected with 70% ethanol and shaved with a disposable razor. Interdigital skin was cut out using a scalpel and forceps and transferred to a 50 mL Falcon tube with a transport medium (DMEM + Penicillin/Streptomycin (100 U/mL/100 μg/mL), Gentamycin (50 µg/mL) and Amphotericin B (2.5 μg/mL)) and incubated for 1 h at room temperature (RT) with occasional shaking. Subsequently, the skin samples were transferred to 6 cm Petri dishes in a sterile biosafety cabinet. The dermal and fatty tissue were removed, as much as possible, and the specimens were cut into smaller cubes (2 × 2 × 2 mm^3^) using a scalpel and forceps. For washing purposes, the tissue cubes were transferred successively to three 6 cm Petri dishes containing Dulbecco’s phosphate buffered saline (DPBS) for 1 min each. Afterwards, the skin pieces were transferred to 15 mL Falcon tubes (2 per donor) with 3 mL of 0.25% Trypsin-EDTA solution (TEDTA) for overnight incubation at 4 °C, then at RT for 1 h with occasional shaking. Foetal calf serum (FCS, 2 mL) was added to stop the enzymatic digestion. The tubes were shaken vigorously, and the cell suspension was microfiltered through cell strainers (pore size Ø100 μm) into 50 mL centrifuge tubes. The microfilter was rinsed twice with DPBS w/o MgCl_2_/CaCl_2_. The tubes were centrifuged at 1500 rpm at RT for 4 min, the supernatant was aspirated, and the cell pellets were re-suspended in 10 mL of the culture medium K-SFM (ThermoFisher Scientific, Loughborough, UK), supplemented with Penicillin/Streptomycin (100 U/mL/100 μg/mL), Gentamicin (50 ug/mL) and Amphotericin B (2.5 µg/mL)), and the cells were seeded into a T75 flask. The flasks were incubated at 37 °C, 5% CO_2_ and 95% relative humidity. The medium was changed every 2 to 3 days (the subsequent medium was supplemented only with Penicillin/Streptomycin after the first two days).

To establish pure cultures of keratinocytes and fibroblasts, respectively, the primary cultures were differentially trypsinised. Under TEDTA-treatment, fibroblasts detach from cell culture dishes after 3 min, whereas keratinocytes detach only after 8 to 15 min. Hence, the primary cultures were washed with DPBS w/o MgCl_2_/CaCl_2_, then incubated with 0.25% TEDTA for 3 min at 37 °C. The enzymatic reaction was stopped with an equal volume of 10% FCS in DPBS w/o MgCl_2_/CaCl_2_. The cell suspension (containing mostly fibroblasts) was centrifuged (1500 rpm, RT, 4 min), and the cells were reseeded into a new T75 flask containing K-SFM + Penicillin/Streptomycin medium. The remaining cells in the first flask (predominantly keratinocytes) were provided with fresh K-SFM medium and further incubated as described above. This procedure was repeated every two days until the keratinocyte cultures were free of fibroblasts (usually 3 or 4 cycles). When the keratinocyte colonies reached confluency, they were sub-cultured using 0.25% TEDTA for 3 min to remove any remaining fibroblasts, then for a further 6 min at 37 °C to lift the keratinocytes and seeded into Nunc six-well plates (ThermoFisher Scientific, Loughborough, UK). The keratinocytes were used for the infection experiments in passages 5–7, fibroblasts were used in passages 3–7.

As the primary cells were naturally infected with *Mycoplasma* ssp., they were treated with Geneflow BIOMYC™ 1 and 2, according to the manufacturer’s instructions. BIOMYC-1 contains tiamutin and is used for 4 days (two media changes), BIOMYC-2 contains minocycline and is used for 3 days (one medium change). The cells were tested for the presence of *Mycoplasma* spp. by PCR before and after treatment.

### 2.2. Immunofluorescent Analysis of Primary Isolated Cells

For the immunocytochemistry, the cells were seeded into 48-well plates and incubated as described above; the cells were fixed with 4% buffered paraformaldehyde (10 min, RT) before confluence was reached. The blocking and permeabilizing steps were combined, incubating the cells with 10% normal goat serum and 0.3% Triton X 100 in DPBS for 30 min. Afterwards, the primary antibody was applied. The dilutions used were 1:50 for Ki67 (rat monoclonal, coupled to FITC; 11-5698-82, eBioscience by Life Technologies, Frankfurt, Germany), 1:500 for Vimentin (mouse monoclonal, coupled to Cy3; C 9080, Sigma Aldrich now Merck, Darmstadt, Germany), 1:100 for CK 14 (guinea pig polyclonal; ABIN 113455, antibodies online, Aachen, Germany) and 1:50 for panCK (mouse monoclonal; DLN-07797, Dianova GmbH, Hamburg, Germany). Incubation ensued in a dark wet chamber for 4 h (RT). The subsequent washing steps (3× DPBS for 5 min each) were followed by incubation with the respective secondary antibody (2 h, RT; goat-anti-mouse-Alexa488 (115-545-062, Dianova, GmbH, Hamburg, Germany) and/or donkey-anti-guinea-pig-Alexa594 (ABIN 611967, antibodies online). The counterstaining was performed using Hoechst 33,342 (1:1000, 10 min, RT; 14533, Sigma Aldrich now Merck, Darmstadt, Germany). All the stainings were accompanied by a secondary antibody control, omitting the primary antibody, to check for unspecific staining. Images were taken using a Nikon TE2000S fluorescent microscope with AR software.

### 2.3. Cell Stimulation

The stimulating bacteria preparations (*D. nodosus* MM261 [6] and *F. necrophorum* DSM 21,784 [20]) were inactivated, either by heat (96 °C for 10 min), UV (20 min), or formalin fixation (4% PFA, for 20 min). Lipopolysaccharide (LPS) from *E. coli* was used as a control (Merck Life Sciences, Dorset, UK). The protein concentration was determined using a Qubit protein kit, then diluted in a culture medium to create aliquots of 10 µg/mL and frozen at −20 °C until required. The skin cells were grown to be 95–100% confluent in Nunc six-well plates (ThermoFisher Scientific, Loughborough, UK) before stimulation. Approximately 2 h before stimulation, the medium was aspirated and the cells were washed with warmed DPBS; the latter was replaced with 900 µL fresh medium per well and then incubated at 37 °C. The stimulating bacteria were diluted to 100 µL aliquots at a final concentration of 1 µg/mL. At time-point zero, inactivated bacteria (or the control medium) were added to each well, mixed gently and returned to 37 °C. After 2 h, 8 h and 24 h of incubation, the aliquots of media were harvested (4 × 240 µL per well) and immediately placed on ice for the subsequent ELISA measurements. Qiagen RA1 buffer (350 µL) was added to the wells to lyse the cells. The wells were scraped using a swivel-blade cell scraper (Greiner Bio-One Ltd., Bristol, UK) to maximise the yield. This solution was collected into a 1.5 mL Eppendorf lock-cap tube and frozen immediately at −20 °C, to be used for the RNA extraction.

### 2.4. RNA Isolation, cDNA Synthesis and RT-qPCR

RNA was isolated using a NucleoSpin RNA isolation kit (Machery-Nagel, Düren, Germany) following the manufacturer’s recommendations. The RNA samples were quantified using the Qubit 3.0 and RNA high-sensitivity dye (Qiagen). The RNA was diluted in water and cDNA was synthesised using M-MLV Reverse Transcriptase (Promega, Madison, WI, USA), according to the manufacturer’s instructions. The final volume of each reaction was diluted in RNAse/DNAse-free water (Fisher Scientific, Loughborough, UK).

The ovine mRNA expression levels of β-Actin (*ACTB*), Cyclophilin (*PPIA* and, selected cytokines IL-1β and CXCL8 were investigated by RT-qPCR (for the primer details, see Table 1). For the qPCR of the target genes, a minimum of three forward and three reverse primers were designed and assessed by RT-qPCR to identify the final primer set (Table 1). All the assays were performed with a manual reaction setup using the BioRad CFX Connect real Time PCR Detection System (BioRad, London, UK). The reactions contained 5 μL of cDNA (1/100 dilutions) in 1× SYBR green qPCR master mix (Sigma-Aldrich, Dorset, UK) with 1μM of forward and reverse primers (Sigma Aldrich, Dorset, UK).

All the standard dilutions, samples and no template controls (NTC) were performed in triplicate. The samples were subjected to an initial denaturation at 95 °C for 10 min, followed by 45 cycles at 95 °C for 10 s, 60 °C for 50 s, 72 °C for 1 min and a final dissociation step at 97 °C.

### 2.5. ELISA

The IL-1β, and CXCL8 proteins were measured from the collected tissue culture supernatants of the cultured fibroblasts and keratinocytes. The IL-1β and CXCL8 were measured using specific sandwich ELISA protocols, as previously described in detail (IL-1β and CXCL8: Haig et al., 1996 [24]), using the antibodies (Table 2). Recombinant ovine IL-1β (Kingfisher^®^ Biotech, Minneapolis, MN, USA) and recombinant ovine CXCL8 (obtained from Moredun Research Institute), produced as described in [25], were used as quantifiable ELISA standards. All the ELISA plates were read on a VarioSkan (ThermoFisher, UK) pathlength corrected for flaws in the plates (650 nm reading subtracted from 450 nm reading), and had the average blank value subtracted before being analysed using PRISM 8.01 (GraphPad Software Inc., San Diego, CA, USA).

### 2.6. Statistical Analysis

The statistical analyses of the cytokine concentrations were calculated using a polynomial quadratic regression in GraphPad Prism version 7b.

## 3. Results

### 3.1. Characterisation of Ovine Interdigital Skin Keratinocytes and Fibroblasts

The cell isolation procedure from the interdigital ovine skin resulted in mixed-cell cultures of keratinocytes and fibroblasts (Figure 1A) that were subsequently purified.

The keratinocytes showed the typical, cobblestone-like morphology of epidermal cells in culture (Figure 1B). In the immunocytochemical analysis, the identity of the keratinocytes was verified using an anti-pan-cytokeratin-antibody (panCK) and the basal cell-layer marker, cytokeratin (CK) 14 (Figure 1D,F). The cultured fibroblasts displayed a spindle shape (Figure 1A) and were positive with an Anti-Vimentin antibody (Figure 1C) but negative with Anti-PanCK (not shown). To prove cell viability, we also stained for the classical proliferation marker, Ki67 (Figure 1E,G), and found the majority of cells actively dividing.

### 3.2. Expression of Pro-Inflammatory Mediators in Primary Ovine Skin Keratinocytes and Fibroblasts in Response to D. nodosus Stimulation

The mRNA transcript levels of IL-1β and CXCL8 were determined in the primary ovine skin keratinocytes and fibroblasts in response to stimulation by *E. coli* LPS, different *D. nodosus* preparations (heat-inactivated, UV-inactivated, formalin-fixed) or heat-inactivated *F. necrophorum* for 2 h, 8 h and 24 h in the presence or absence of a *M. fermentans*. infection.

As expected, using primary cells isolated from different individual sheep, the variability in the targeted gene expression between the batches was high for both cell types. In the keratinocytes, free of *M. fermentans*, little differences were observed at 2 h and 8 h of stimulation (Figure 2). At 24 h, increased transcript levels were observed in both targets (IL-1β and CXCL8) in response to stimulation with LPS (81-fold, *p* = 0.0076), different *D. nodosus* preparations (32–102-fold, *p* = −0.0051 − 0.34) and heat-inactivated *F. necrophorum* (83-fold, *p* = 0.0072; Figure 2C). In the fibroblasts, the CXCL8 transcript levels increased 18-fold (*p* = 0.018) in response to the LPS and 42-fold (*p* = 0.0005) in response to the heat-inactivated *D. nodosus* after 2 h, and 25-fold (*p* = 0.0124) to the LPS and 86-fold (*p* = 0.006) to the heat-inactivated *D. nodosus* after 8 h (Figure 3A,B). The IL-1β transcript levels increased 41-fold (*p* = 0.03) in response to the LPS, 74-fold (*p* = 0.05) in response to the heat-inactivated and 39-fold (*p* = 0.05) in response to UV-inactivated *D. nodosus* after 2 h (Figure 4A), and 30-fold (*p* = 0.039) in response to the heat-inactivated *D. nodosus* after 8 h (Figure 4B). As the ovine interdigital skin cells were naturally infected by *M. fermentans* prior to the cell isolation, we investigated the expression of IL-1β and CXCL8 in response to the same stimuli (Figure 2, Figure 3, Figure 4 and Figure 5, right columns). No increases in transcript levels were observed in the keratinocytes naturally infected with *the M. fermentans* (Figure 2D–F). This was similar for the *M. fermentans*-infected fibroblasts (Figure 3D–F and Figure 4D,F), except for IL-1β expression, which was increased 30-fold (*p* = 0.0002) after 8 h of stimulation with LPS and 20-fold (*p* = 0.04) after 8 h of stimulation with UV-inactivated *D. nodosus*, respectively (Figure 4E).

To investigate if those transcript changes resulted in the release of pro-inflammatory mediators (protein), we measured the protein levels of IL-1β and CXCL8, by ELISA, in culture supernatant collected from the same stimulation experiments, used to measure the transcript levels. No release of IL-1β was detected in any of the keratinocyte or fibroblast culture supernatants. In both naturally *M. fermentans*-infected, as well as *M. fermentans*-free keratinocytes, very little or no release of CXCL8 was detected in response to stimulation. In the *M. fermentans*-free fibroblasts, a consistently higher release of CXCL8 was observed in response to the LPS, heat-inactivated and UV-inactivated *D. nodosus* compared to the control cells (Figure 5C). However, a significant increase in the release of CXCL8 was only stimulated by the LPS and UV-inactivated *D. nodosus* (Figure 5C). In the fibroblasts naturally infected with *M. fermentans*, no increase in protein release through stimulation was observed for any of the four pro-inflammatory mediators measured (Figure 5D).

In summary, the keratinocytes and fibroblasts naturally infected with *M. fermentans* showed little to no response to LPS, a range of *D. nodosus* preparations or heat-inactivated *F. necrophorum*. While in the fibroblasts (infected and *Mycoplasma* spp. free), the CXCL8 mRNA expression was generally mirrored in the CXCL8 release; no IL-1β release was detected despite increases in the IL-1β mRNA. This potentially suggests the signal for intracellular pre-IL-1β processing may be lacking when culturing primary cell types individually.

## 4. Discussion

### 4.1. Primary Ovine Interdigital Skin Cell Cultures as a Model for Infection

Although footrot was described more than 80 years ago and its impact on animal welfare and economy is well recognized [26], surprisingly, only one report of footrot ovine skin cell cultures is found in the literature [18]. This is astonishing since 2D cell cultures of human and animal origin derived from many different types of tissue are very well reported and widely used in basic research [27], and even more, since protocols of cell isolation and maintenance in culture are numerously available for ovine cells and tissues as well, including skin and epithelial cells [28,29,30]. However, this might be explained with the fastidious nature of the causative agents of ovine footrot and the unclear role of each of the suspected microbes involved.

The most important advantage of primary cell cultures consisting of a single cell type is that they allow for the basic study of the microbial interaction with this precise cell type without the influence of other cells or the immune system, as described, e.g., for viral agents of ovine skin diseases, as well [31,32]. Primary cells may be more fastidious in obtaining and maintenance and also more heterogeneous in morphology and physiology compared to cell lines [33]. However, the former might also be seen as an advantage as they reflect inter-individual variations. Additionally, the phenotype and responsiveness to external stimuli may be altered in cell lines due to genetic differences/manipulation(s) and serial passaging for a long time. Thus, the results obtained from cell lines do not necessarily reflect naturally occurring phenomena, and comparative experiments using primary cells are advised [33,34,35]. For certain tissues of livestock, this might even be achieved easily and without raising ethical concerns in terms of animal experiments since source material can be collected from local abattoirs, thereby also adhering to the 3R principles.

There are many protocols for the isolation of skin cells from biopsies; an enzymatic treatment using trypsin or dispase is usually recommended to extract keratinocytes and fibroblasts from skin tissue and obtain individual cell cultures [27]. The procedure described here to separate the two cell types and purify each culture (differential trypsinisation) was also successfully employed for bovine skin cells [36]. As seen with our keratinocytes, Watkins et al. [28] describe the cobblestone-like appearance of ovine skin keratinocytes after enzymatic tissue digestion and cell adhesion to the culture vessel. However, they confirmed the cell identity with an Anti-Involucrin immunofluorescence staining, which, indeed, proves the cells as being keratinocytes as well as their late differentiation state. Being CK14 positive, the early differentiation state of our primary isolations is shown, which is favourable in terms of long-lasting 2D cell cultures [37]. The ovine interdigital skin fibroblasts displayed the typical cell morphology (spindle shape, in high confluency, similar to a school of fish) and stained positive for the intermediate filament vimentin being an important compound of the cytoskeleton of mesenchymal cells. However, strongly proliferating cultured keratinocytes may not be as easily identified by morphology and were shown to express vimentin as well [38]. Therefore, our fibroblast cultures were also subjected to Anti-PanCK-stainings to confirm the absence of cytokeratins, as another proof of cell identity [35,39].

### 4.2. Response of Ovine Interdigital Skin Fibroblasts and Keratinocytes to Microbial Preparations

The stimulation with LPS, different *D. nodosus* preparations or *F. necrophorum* resulted in higher expression levels and altered release of different pro-inflammatory mediators by the primary ovine keratinocytes and fibroblasts. We had shown previously that fibroblasts from other skin areas showed increased expression of IL-1β mRNA in response to stimulation with heat-inactivated *D. nodosus,* demonstrating those bacteria can elicit a pro-inflammatory response in the absence of active infection [18]. Increased expression of IL-1β and CXCL8 mRNA have also been observed in bovine digital dermatitis lesions, caused by treponemes [40]. In the ex vivo organ culture (EVOC) model of ovine interdigital skin, infection with *D. nodosus* stimulated IL-1β release; however, CXCL8 levels increased in both infected and mock-infected explants [19]. In the herein used cell cultures, in the fibroblasts, the CXCL8 mRNA expression was generally mirrored in the CXCL8 release; however, no IL-1β release was detected in the keratinocytes or fibroblasts, despite increases in the IL-1β mRNA. Taken together, these suggest the signal for intracellular pre-IL-1β processing may require the interaction of different cell types, the 3D skin structure, or an infection with live *D. nodosus*.

The keratinocytes and fibroblasts naturally infected with *M. fermentans* showed little response to LPS, a range of *D. nodosus* preparations or heat-inactivated *F. necrophorum*. This is not surprising, as chronic infections of monocytes and macrophages with intracellular low-pathogenic *Mycoplasma* spp. have been shown to impair their inflammatory response to live bacteria and bacterial products [41,42]. While a Mycoplasma infection in primary cells often is a contamination-altering cytokine expression pattern, signal transduction and cellular metabolism [43], in the context of footrot, a polymicrobial infection in which *M. fermentans* is significantly associated with the disease state [7], this impaired response of the cells to stimulation is an interesting finding because it might be an important factor during the disease establishment in vivo, as well. This suggests that naturally *M. fermentans*-infected primary cells of the ovine interdigital skin could be an appropriate model to investigate complex polymicrobial host pathogen interactions in the context of ovine footrot.

### 4.3. Limitations

Single cell line stimulation studies do suffer from some minor limitations. As there are no interactions between different cell types, a lack of a 3D structure, no air interface and a limited life span of the cell lines, the conclusions need to be considered in that context and as a model to further build on. However, as co-culture infection studies in anaerobic environments are difficult to achieve, this study highlights some potential important avenues of investigation for footrot prevention

## 5. Conclusions

We show that primary single cell culture models complement EVOC models to study different aspects of the host response to *D. nodosus*. Furthermore, the natural cellular co-infection with Mycoplasma gives important insight into the impact of multiple pathogens being present in the host response in the primary target cells.

## Figures and Tables

**Figure 1 animals-12-03235-f001:**
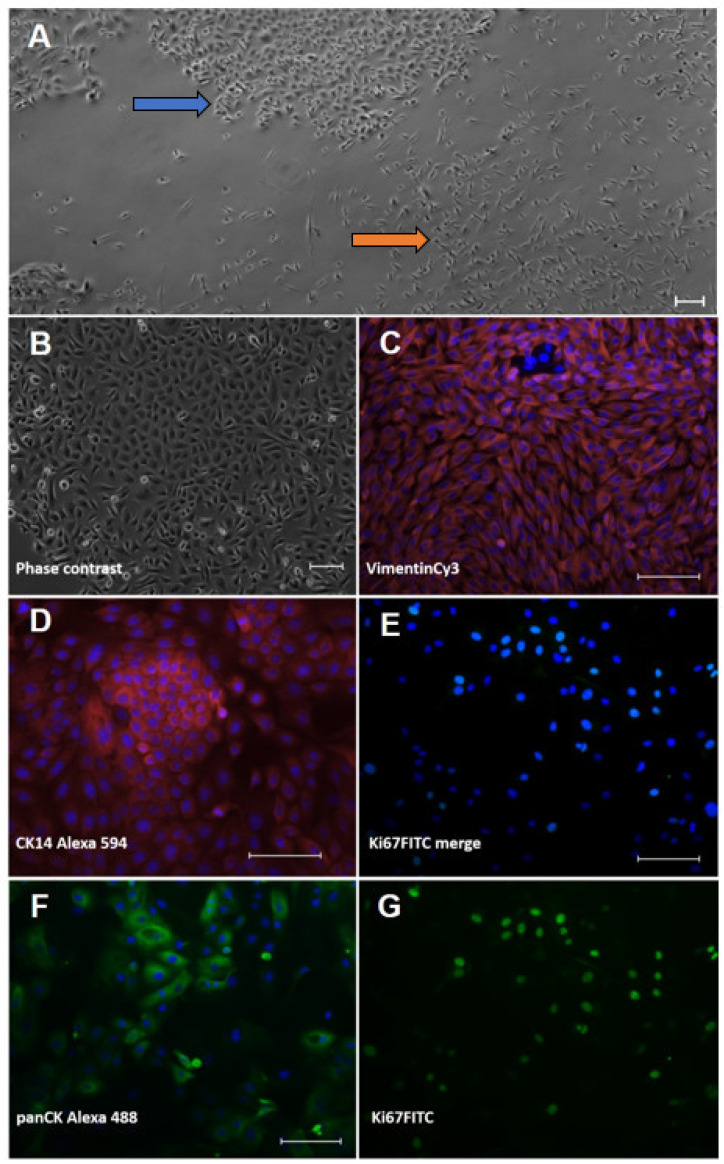
Characterisation of isolated ovine skin cells using immunofluorescence staining. Keratinocytes (left column; (**B**,**D**,**F**)) and fibroblasts (right column; (**C**,**E**,**G**)) were isolated from ovine interdigital skin and displayed typical cell morphologies in culture. (**A**) primary isolation showing mixed islets of cobblestone-like keratinocytes (blue arrow) and spindle-shaped fibroblasts (orange arrow). (**B**) purified keratinocyte culture. Cell types were stained for typical lineage markers, i.e., (**D**) cytokeratin (CK) 14 and (**F**) pan-cytokeratin in keratinocytes and (**C**) vimentin in fibroblasts, respectively. Proliferation was shown using an antibody directed against the typical proliferation marker, Ki67 (**E**) merged picture and the extracted FITC channel in (**G**)). The nuclear counterstain ensued with bisBenzimide Hoechst 33342; all staining was accompanied by a secondary antibody control (omission of first antibody). All scale bars represent 100 µm.

**Figure 2 animals-12-03235-f002:**
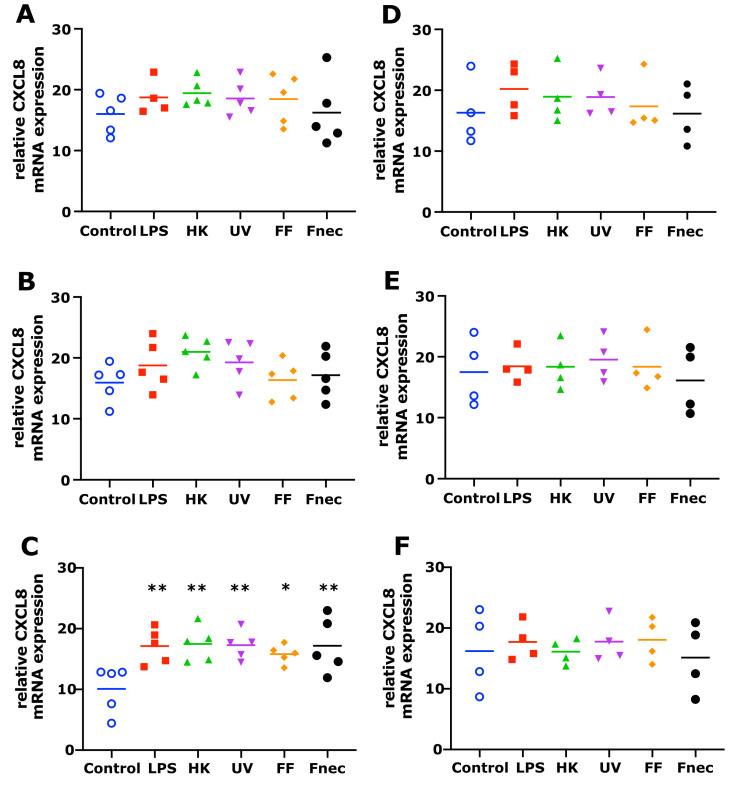
CXCL8 mRNA expression in interdigital ovine keratinocytes. Primary keratinocytes free of *M. fermentans* (*n* = 5, (**A**–**C**)) or naturally infected with *M. fermentans* (*n* = 4, (**D**–**F**)) were stimulated with 1 µg/mL *E. coli* LPS, heat-inactivated *D. nodosus* (HK), UV-inactivated *D. nodosus* (UV), formalin-fixed *D. nodosus* (FF) or heat-inactivated *F. necrophorum* (Fnec) (cell culture medium was used as diluent and control) for 2 h (**A**,**D**), 8 h (**B**,**E**) and 24 h (**C**,**F**). Horizontal line indicates the mean. Each replicated symbol represents a different cell line from an individual sheep. One-way-ANOVA, followed by Dunnett’s multiple comparisons test: * *p* ≤ 0.05, ** *p* ≤ 0.01.

**Figure 3 animals-12-03235-f003:**
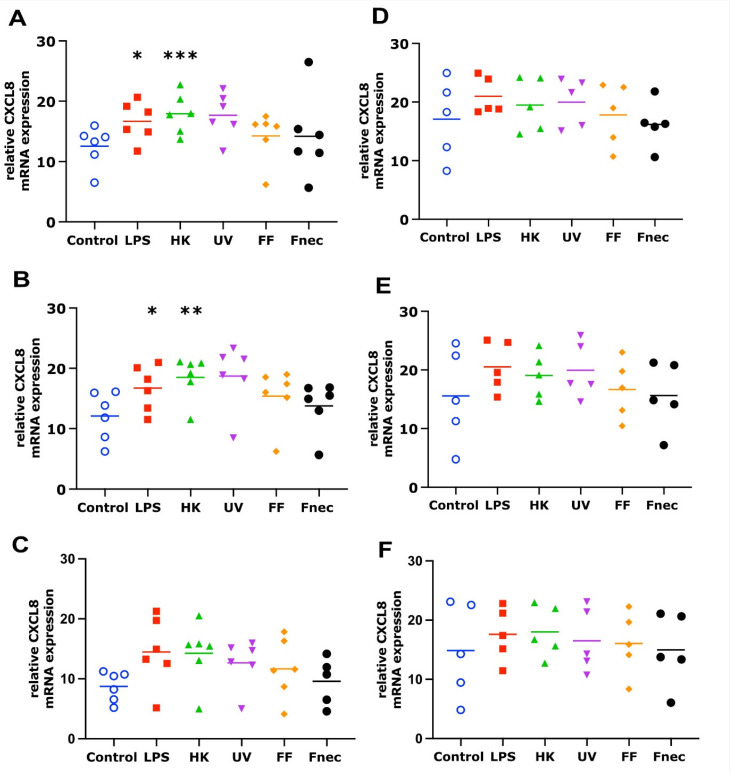
CXCL8 mRNA expression in interdigital ovine fibroblasts. Primary fibroblasts free of *M. fermentans* (*n* = 6, (**A**–**C**)) or naturally infected with *M. fermentans* (*n* = 5, (**D**–**F**)) were stimulated with 1 µg/mL *E. coli* LPS, heat-inactivated *D. nodosus* (HK), UV-inactivated *D. nodosus* (UV), formalin-fixed *D. nodosus* (FF) or heat-inactivated *F. necrophorum* (Fnec) (cell culture medium was used as diluent and control) for 2 h (**A**,**D**), 8 h (**B**,**E**) and 24 h (**C**,**F**). Horizontal line indicates the mean. Each replicated symbol represents a different cell line from an individual sheep. One-way-ANOVA, followed by Dunnett’s multiple comparisons test: * *p* ≤ 0.05, ** *p* ≤ 0.01, *** *p* ≤ 0.001.

**Figure 4 animals-12-03235-f004:**
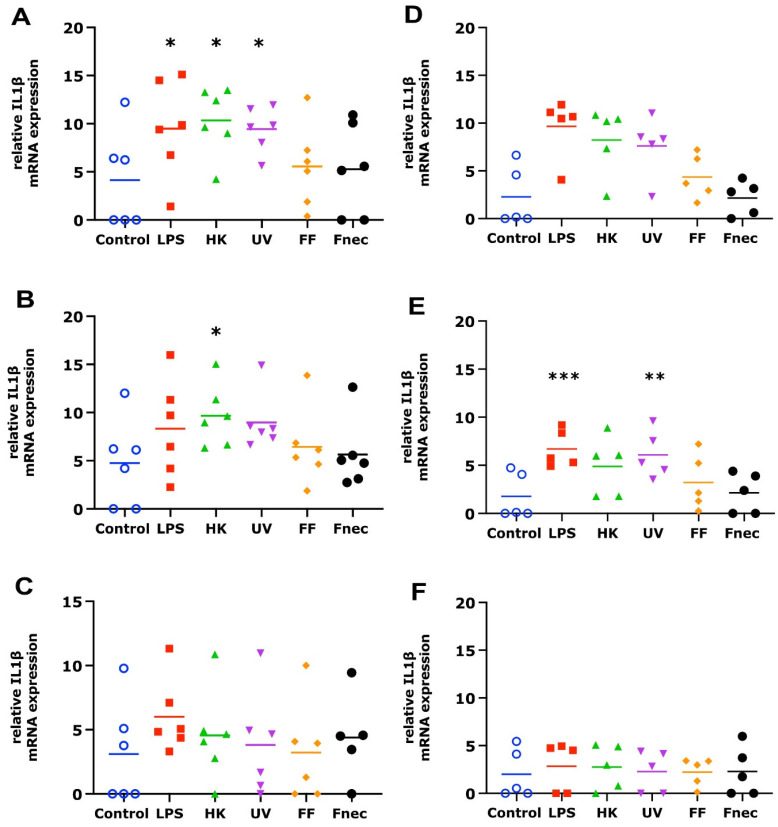
IL-1β mRNA expression in interdigital ovine fibroblasts. Primary fibroblasts free of *M. fermentans* (*n* = 6, (**A**–**C**)) or naturally infected with *M. fermentans* (*n* = 5, (**D**–**F**)) were stimulated with 1 µg/mL *E. coli* LPS, heat-inactivated *D. nodosus* (HK), UV-inactivated *D. nodosus* (UV), formalin-fixed *D. nodosus* (FF) or heat-inactivated *F. necrophorum* (Fnec) (cell culture medium was used as diluent and control) for 2 h (**A**,**D**), 8 h (**B**,**E**) and 24 h (**C**,F). Horizontal line indicates the mean. Each replicated symbol represents a different cell line from an individual sheep. One-way-ANOVA, followed by Dunnett’s multiple comparisons test: * *p* < 0.05, ** *p* < 0.01, *** *p* ≤ 0.001.

**Figure 5 animals-12-03235-f005:**
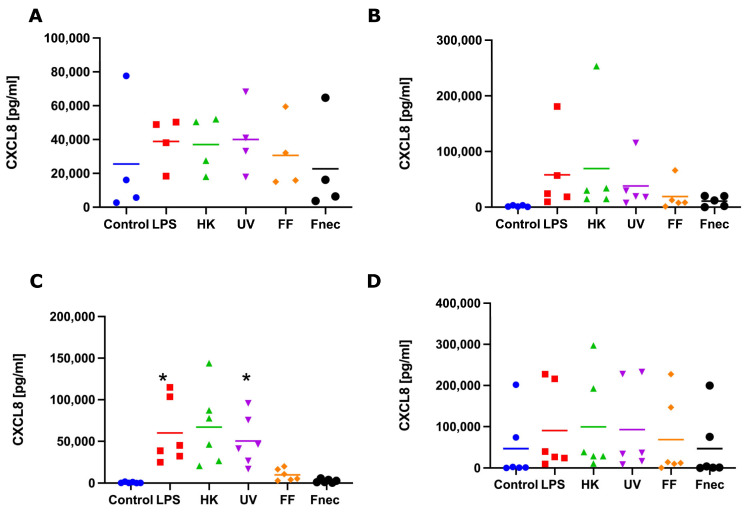
CXCL8 protein released into cell culture medium by interdigital ovine keratinocytes and fibroblasts. Primary keratinocytes free of *M. fermentans* (*n* = 4, (**A**)) or naturally infected with *M. fermentans* (*n* = 5, (**B**)) and primary fibroblasts free of *M. fermentans* (*n* = 6, (**C**)) or naturally infected with *M. fermentans* (*n* = 6, (**D**)) were stimulated with 1µg/mL *E. coli* LPS, heat-inactivated *D. nodosus* (HK), UV-inactivated *D. nodosus* (UV), formalin-fixed *D. nodosus* (FF) or heat-inactivated *F. necrophorum* (Fnec) for 24 h (cell culture medium was used as diluent and control). Horizontal line indicates the mean. Each replicated symbol represents a different cell line from an individual sheep. One-way-ANOVA, followed by Dunnett’s multiple comparisons test: * *p* < 0.05.

**Table 1 animals-12-03235-t001:** Primers used in the qPCR reactions.

Target	Primer	Size	GenBank ID	Source
Ov_Actin_F	TGTGCGTGACATCAAGGAGAA	67 bp	AF129289	[21]
Ov_Actin_R	CGCAGTGGCCATCTCCTG			
Ov_PPIA_F	TGAGCACTGGAGAGAAAGGATTTG	84 bp	AY251270	[22]
Ov_PPIA_R	AGTCACCACCCTGGCACATAA			
Ov_IL1 β _F	TTCTGCATGAGCTTCGTACAA	115 bp	X54796	[23]
Ov_IL1 β _R	GGGTCGGTGTATCACCTTTTT			
Ov_CXCL8_F	GAGAAGTCCTCTGGGACAGC	102 bp	NM_001009401.1	[15]
Ov_CXCL8_R	CAGCCAGCTTGGAAGTCATA			

**Table 2 animals-12-03235-t002:** ELISA antibodies and dilutions for each protein target.

Target	Capture Antibody	Protein Standard	Secondary Antibody	Tertiary Antibody	Standard Curve Range
IL-1β	Bio-Rad Mouse anti Ovine IL-1β clone 1D4 (MCA1658, 1:200)	Ovine IL-1β RP0656V-005	rabbit anti sheep IL-1β polyclonal (AHP423, 1:500)	Dako (PO448) Goat anti rabbit-HRP conjugate (1:500)	375–30,000 pg/mL
CXCL8	Bio-Rad Mouse anti Ovine IL-8 clone 8M6 (MCA1660, 1:200)	Kingfisher Biotech Ovine & Caprine IL-8 (CXCL8) RP0488V-005	Bio-Rad rabit anti-sheep IL-8 polyclonal (AHP425, 1:500)	Dako (PO448) goat anti rabbit-HRP (1:500)	25.6–1640 pg/mL

## Data Availability

Data available upon request to corresponding author.

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
