# Peer review of "Natural Mycoplasma Infection Reduces Expression of Pro-Inflammatory Cytokines in Response to Ovine Footrot Pathogens"

_animals, 2022, doi:10.3390/ani12233235_

Round 1
Reviewer 1 Report
The paper from Blanchard et al. is a relatively straightforward in vitro study of the effects of treatment of sheep keratinocytes and fibroblasts with inactivated bacterial extracts from D. nodosus (DN), F necrophorum (FN) with and without Mycoplasma infection. Overall, the paper is descriptive and does not provide any mechanistic data. It can be improved, in my opinion, as follows:
1. There is no mention of FN in the introduction. This needs to be corrected in the context of footrot infection.
2. The methodology needs more details as follows
a) add a picture of the distal limb and the process to produce the cells. This would be useful to the reader uninformed of anatomy of foot.
b) There is no method for Mycoplasma growth and infection of the cell cultures, and no data to demonstrate the absence in cell cultures of Mycoplasma by PCR (show data supplementary).
c) Why no stimulation with live bacteria (except the Mycoplasma); clarify.
d) Statistics P values <0.5 significant? Clarify in 2.6.
Results
a) Fig 1 needs arrows to point out the keratinocytes and fibroblasts, and the negative (no primary antibody) controls (could go in suppl).
b) Measuring proliferation -usually one would use an in vitro assay to quantify cell growth, there are plenty about. The figure does not give a real measure of proliferation.
c) In the graphs, how many different sheep cell cultures were established? Does each symbol represent an individual cell line from an individual sheep?
d) No details have been provided of the concentrations of bacterial extracts used to infect the cells, or indeed how many live Mycoplasma are used.
e) What happens if you infect with a mixture of FN and DN bacteria with Mycoplasma? Surely this would be a better mimic of what might be happening in vivo?
f) How long after you remove the Mycoplasma do you then infect the cells? Do you think prior Mycoplasma infection has induced a tolerance?
Discussion
I don't really get a good idea of the significance of the data biologically in the discussion. Also, i am unclear of the current understanding of the pathobiology of infection that causes footrot and this ought to be discussed, perhaps as a working model.
As it's a multi-organism disease, so does FN cause skin damage to allow DN to invade, or does Mycoplasma infection dampen the immune response, allowing FN to breach skin barrier and allow DN to invade?
I think a section describing the limitation of the study would be useful and what else could the models be used for? What about live infection, is an anaerobic environment difficult to produce?
Reviewer 2 Report
There is growing evidence that mycoplasmas contribute to increased severity of disease, this work could make an important contribution to the growing knowledge-base in this area.
A major assumption made as part of the experimental work presented is that Mycoplasmopsis fermentans is naturally infecting the primary cell lines described in the manuscript. The manuscript makes reference Infect. Immun. 2021;89 as evidence that the cell lines are infected with M. ferementans; are these the skin biopsies from which the cell lines were derived? Are there other mycoplasmas that might also be present? In order to satisfy Koch's postulates it is essential that the effects observed in the naturally infected cell lines is observed in 'sterile' cell lines that have been reinfected with M. ferementans. An isolate would be useful for this. An M. ferementans MAG (from one of your cell lines) could be used to confirm the culture purity of a filtrate that might be used to reinfect (if an isolate can't be obtained).
Other more minor items:
There are several very long sentences! e.g. To further understand... [line 35] ...protein release is observed [line 42]. Please revise.
There are no details about the bacterial strains used for the preparation of the LPS and inactivated materials. please include.
There seems to be too much detail about the methods used to establish the primary cell lines. Reference to previous papers establishing cell lines and including only the key details of your methods is recommended.
Round 2
Reviewer 1 Report
No further comments. Questions answered; obviously a multi-organism disease, inc Mycoplasma. perhaps in future, the pathogenesis might be unravelled.